# Compound Castings for the Coke Industry

**DOI:** 10.3390/ma17143539

**Published:** 2024-07-17

**Authors:** Tomasz Wróbel

**Affiliations:** Department of Foundry Engineering, Silesian University of Technology, 7 Towarowa Street, 44-100 Gliwice, Poland; tomasz.wrobel@polsl.pl

**Keywords:** compound casting, steel, cast iron, coke

## Abstract

In this paper, issues related to the technology of compound castings composed of two parts, i.e., the working layer and the supporting part, made of X46Cr13 high-chromium steel and EN-GJL-HB 255 grey cast iron, respectively, in a liquid–solid system by pre-installing a monolithic insert in the mould cavity are presented. As a part of the research, the mechanism of formation of transitional zones in the bonding area of the above-mentioned two alloys was identified and described. It was shown that the phenomenon that determines the formation of a permanent bond between the joined materials is the transport of C and heat from the “high-carbon and hot” material of the supporting part poured into the mould in the form of liquid cast iron to the “low-carbon and cold” material of the working layer placed in the form of a steel monolithic insert inside the mould cavity. In the paper, the suitability of the compound castings technology developed for use in the coke industry is also presented. Full-size high-chromium steel–grey cast iron compound casting plates designed for the coke quenching car lining were positively verified in real coke plant operating conditions.

## 1. Introduction

In recent years, the demand for special-performance castings in various industries has been increasing, including those featuring high hardness, high resistance to abrasive wear, and corrosion resistance at room or elevated temperatures. Such parts are frequently made entirely of expensive, and hard-to-find materials, e.g., high-alloy steels and cast steels, alloys with Ni or Ti matrixes, and others [1]. In many cases, this is unnecessary because the high usable properties are often required not for the entire casting but only for its working surface. In such cases, the operating conditions of the casting result in wearing only one surface, the so-called working surface, which leads to its permissible dimensional tolerance being exceeded and forces the entire casting to be scrapped, which, taking into account the use of expensive materials, can be considered uneconomical.

Considering the above-mentioned factors, the technology of compound castings becomes particularly important. It can be used when, as described above, the high usable properties criterion is required only for the outside working layer of a casting, while its remaining part plays only the role of a supporting structure, not exposed to direct factors resulting in, e.g., corrosive or abrasive wear [2,3,4,5,6,7,8,9,10,11,12,13,14,15,16,17,18,19,20]. Furthermore, this technology is one of the most economical ways to improve the working surface of castings as it allows the production of compound components directly during the manufacturing process. As a result, for massive castings with simple geometry, it can be a significant technological alternative for the commonly used surfacing and thermal spraying or surface heat treatment. In addition, using the technology of compound casting, apart from its economic advantages and unlike the above-mentioned technologies of surfacing and surface heat treatment, does not involve the risk of cracking in the heat-affected zone resulting from the use of the above-mentioned methods [21,22,23,24].

The production of compound castings composed of two basic components, i.e., working part (layer) and supporting part, also referred to as bi-metal castings, is typically carried out using three basic technologies, taking into account the state of matter in which the materials are at the time of bonding, i.e., in the following systems:liquid–liquid [2,3,4,5];solid–solid [6,7,8,9];liquid–solid [10,11,12,13,14,15,16,17,18,19,20].

For the first technology mentioned, both bi-metal parts consist of materials made exclusively by casting. An example of this technology is one in which two independent gating systems are made that guarantee two-stage filling of the sand mould cavity. According to this manufacturing method, the bimetallic elements of the hammer [2,3], lining of mills ball [4], or rolls [5] are cast in material configurations of the chromium cast iron working layer with a low-carbon cast steel supporting part. 

In contrast, in solid–solid technology, it is possible to bond the previously produced cast alloys, i.e., grey cast iron with flake graphite with nodular cast iron [6] or cast alloys, with non-cast alloys, e.g., the chromium cast iron with carbon steel [7]. Moreover, with this technology, it is possible to obtain a bond between two steels, e.g., carbon steel and stainless steel, as the paper [8] presented. According to [6,7,8,9], solid–solid technology is limited to small elements with maximal contact surface between both components of approx. 200 mm^2^ and 30 mm in thickness. Additionally, this technology needs an elevated temperature of approx. 1000 °C, pressure at the time of bonding of 1–10 MPa, and a vacuum at the time of bond creation [6,7,8,9]. 

Meanwhile, liquid–solid technology can use a combination of cast and non-cast alloys. In this manufacturing method, the element that enriches the surface of the casting is placed in the mould in the form of a granular [10,11,12] or monolithic [13,14,15,16,17,18,19,20] insert directly before the molten alloy is poured. With the use of granular inserts, compound castings were made in a material configuration working surface layer from WC [10], Al_2_O_3_ [11], or SiC [12] with an unalloyed cast steel or grey cast iron supporting part. This type of compound casting is characterized by a hard working surface layer that is resistant to abrasive wear, which is a decisive factor in its applicability in the mining industry. An example of a technology using the monolithic insert is compound castings in a configuration which comprises hard and resistant-to-abrasive-wear chromium cast iron connected with a weldable low-carbon steel plate [13,14]. Examples of small test castings manufactured with high-chromium or chromium–nickel stainless steel monolithic inserts using liquid–solid technology with grey cast iron are presented in papers [15,16,17,18,19]. It is possible to use 3D-printed elements as monolithic inserts, which was shown in paper [20] in compound casting in a configuration of pure Ti with grey cast iron. 

Another division can be made by considering the nature of the bond of both bi-metal parts. It can be diffusion-based, adhesion-based, or mechanical. Considering the strength of the bond and the ability to dissipate heat, and therefore the service life of the compound casting, the most advantageous bond is obviously the diffusion-based one whose characteristic feature is the formation of a smaller or larger transition zone resulting from physical and chemical phenomena occurring at the layer border: working layer–supporting part. However, the phenomenon of diffusion does not occur for the adhesion-based bond, formed as a result of mutual intermolecular attraction. On the other hand, to obtain a mechanical bond by using the liquid–solid technology, it is necessary to make cuts, protrusions, or hooks in the poured component to connect it to the part formed after the solidification of a liquid alloy poured into the mould. This solution utilises the mechanical bond formed by the phenomenon of casting shrinkage that occurs in the cast-on part, as presented in papers [13,14]. 

As a result, in the paper the results of research on the compound casting technology in the liquid–solid system, based on the method of preparing the mould cavity by pre-installing the monolithic insert of high-chromium steel plate poured with grey cast iron, whose goal was to develop a completely novel and previously undescribed in the literature technologically useful bi-metal plates for use in lining a coke quenching car, are presented. Moreover, the studies aimed to describe the mechanism of the bond in analysed compound castings in which the improvement in abrasive wear resistance is obtained by the application of high-chromium steel and, as a result, the creation of a hard microstructure containing Cr(Fe) carbides in a martensitic matrix, with a simultaneous improvement in corrosion resistance i.e., heat resistance and resistance to the corrosive effect of cooling water which is obtained by the creation of a passive surface layer with Cr oxides.

## 2. Materials and Methods

In reference to the aim of the paper, in general, the coke quenching car is lined with 376 plates with the following dimensions: width 520 mm, length 575 mm, and thickness 25 mm. They are used in the coke production cycle that includes the following steps:Discharging from the bank of coke-ovens onto the quenching car of between a few and more than ten tonnes of coke heated up to 900–1100 °C from a height ranging from 2 to 4 m (Figure 1a),Carrying heated coke to a quenching tower (Figure 1b),Quenching the coke using the dry method with neutral gases or using the wet method with water flowing in a closed circuit that contains lots of harmful corrosion-accelerating pollutants (Figure 1c,d),Dumping the cooled down coke from the quenching car surface onto a chute from where it is carried to the sorting plant.

The above-presented coke production cycle is carried out from 50 to 150 times a day, resulting in the exposure of quenching car liner plates to high temperatures, thermal shocks, the corrosive effect of cooling water, and abrasive wear in the metal–coke system. As a result, there is a problem of unsatisfactory service life of the EN-GJN-HB 340 mottled cast iron plates with Cr ≤ 2 wt.% added used for the lining plates (Figure 2). Such cast iron provides a service life for the lining plates that does not exceed approx. 50% of the total service life designed for the quenching car. However, due to the difficulties encountered while dismantling the heavily corroded screw connections attaching the lining plates to the car surface and the resulting downtime, it is allowed—despite the damage—for them to continue their operation, at the same time creating conditions conducive to corrosive destruction of the remaining car structure.

As a part of the experiment conducted and presented herein, compound castings were carried out using the following procedure: the type and arrangement of the casting gating system was developed using the liquid–solid system with a monolithic X46Cr13 steel insert (Table 1) 5 mm thick (solid/casting working layer) poured with grey cast iron (liquid/casting supporting part) grade EN-GJL-HB 255 (Table 1). Next, a wooden casting model was made, divided into a working layer and a supporting part with the dimensions of the compound casting plate used to line the coke quenching car: width 520 mm, length 575 mm and thickness 25 mm, and making a two-part mould of classic moulding sand with a quartz matrix with bentonite as a binder, with a division plane corresponding to the working layer contact surface with the supporting part. Then, the mould was dried at 180 °C. Subsequently, the monolithic insert contact surface was prepared by sandblasting and coating it with a flux–water solution of Na_2_B_4_O_7_ and H_3_BO_3_, then drying at 120 °C for 1 h to remove H_2_O. Finally, the mould was dismantled and the previously prepared monolithic insert placed inside its cavity (Figure 3).

In the next stage, the mould was assembled and prepared to pour with a liquid cast iron. Then, the metal charge was prepared, allowing us to obtain EN-GJL-HB 255 grey cast iron with flake graphite in a pearlite matrix. After this, the chemical composition of the liquid alloy was checked and corrected after melting the metal charge, if necessary. According to presented procedure, the castings of 8 compound casting plates were carried out; 4 moulds were poured with liquid cast iron at 1450 °C while 4 at 1550 °C. In all 8 cases, the temperature of the monolithic insert inside the mould before its filling was about 20 °C. For each test casting, free air cooling was applied in its mould after pouring. Once cooled down, the casting was knocked out of the mould and the components of the gating system were removed. In the next step, assessment of the quality of the plate compound casting was carried out by visual inspection and non-destructive ultrasonic testing of the area where the working layer connects with the supporting part. The tests were conducted by using a DIO 1000 STARMANS ELECTRONICS flaw detector (STARMANS ELECTRONICS S.R.O., Prague, Czech Republic)with a flat-head PN10-4C, while the area showing proper, permanent bonding of the working layer with the supporting part was considered to be the area where the bottom echo was higher than the echo of the transition zone (the head applied from the side of the working layer (steel plate)).

The scope of laboratory tests included, on the samples cut from the centre of the compound casting, testing of the chemical composition of steel X46Cr13 and grey cast iron by using a LECO GDS500A glow discharge spectrometer(LECO Corporation, St. Joseph, MI, USA). Moreover, in the case of C determination in grey cast iron, an IR analyser (infrared spectroscopy) LECO CS-125 (LECO Corporation, St. Joseph, MI, USA) was used. Additionally, for the samples cut from the centre of the compound casting metallographic testing was carried out using a NIKON Eclipse LV150N (NIKON Metrology Europe NV, Leuven, Belgium) (LOM) optical microscope, a scanning electron microscope (SEM) INSPECT F (FEI Technologies Inc., Hillsboro, OR, USA) with an energy X-ray dispersive spectrometer (EDS), and a transmission electron microscope (TEM) FEI TITAN 80–300 (FEI Technologies Inc., Hillsboro, OR, USA). Samples for LOM and SEM testing were prepared by cutting, grinding, and polishing with the water-based suspension of Al_2_O_3_ and electrolytic etching using LectroPol-5 STRUERS (STRUERS ApS, Ballerup, Denmark)and reagent with the following chemical composition: 3 g FeCl_3_ + 10 cm^3^ HCl + 90 cm^3^ C_2_H_5_OH, with a voltage of 15 V and time of 15 s. Samples for the TEM testing were obtained using the FEI FIB Quanta 200i system (FEI Technologies Inc., Hillsboro, OR, USA)that made it possible to collect lamellas with dimensions of 20 × 8 μm and a thickness of 50 nm using a convergent beam of gallium ions from SEM images. Metallographic tests were complemented by microhardness measurements at three points for each microstructure zone (the distance between points was 2 mm) using the Vickers hardness test with a FUTURE-TECH FM 700 tester (FUTURE-TECH CORP., Fujisaki, Japan) and a diamond indenter (right quadrangular pyramid with a vertex angle of 136°) loaded with a force of 4.9 N (μHV 0.5). Surface hardness measurements at nine points distributed on the samples cut from the centre, half-length, and near-edge areas of the working layer (three measurement points per sample) were made using the Vickers method with a universal SUNPOC SBRV-100 D hardness tester (Guizhou Sunpoc Tech Industry Co., Ltd., Guiyang, China) and a diamond indenter (right quadrangular pyramid with a vertex angle of 136°) loaded with a force of 50 N (HV5).

To determine suitability for industrial applications, i.e., for use of the compound casting plates made in the system as a lining for a coke quenching car, the working surface of the X46Cr13 high-chromium alloy steel and the supporting part made of EN-GJL-HB 255 grey cast iron, a series of tests was carried out covering the following activities:Installing 3 compound casting plates and 3 control plates of cast iron connected with EN-GJN-HB 340 low-alloy mottled cast iron with an average surface hardness of 350HV5 in a coke quenching car (Figure 4),Using plate castings in the operating conditions of a coke quenching car (Figure 5), including from 72 to 81 coke production cycles (discharge, transport, wet quenching, and dumping coke into a chute) per day, 7 days a week,Dismantling of plate castings after the completion of approx. 18,500 coke production cycles for 8 months.

In turn, the scope of industrial tests included the following:Volumetric loss by measuring the thickness, with an accuracy of 0.1 mm, and weight loss by measuring the mass, with an accuracy of 0.1 kg, of the compound plates under the operating conditions of a coke quenching car and comparing the results with the damage of uniform EN-GJN-HB 340 mottled cast iron plates (Table 1) operated in the same conditions,Testing the microstructure of the compound casting plates after operation using LOM (the samples cut from the centre of compound casting),Hardness tests after operation of the compound casting according to the above-mentioned procedure using the Vickers method.

## 3. Results and Discussion

Figure 6a,b present a view of an example compound casting plate in the following arrangement: working layer of X46Cr13 alloy steel and the supporting part made of EN-GJL-HB 255 grey cast iron. The conducted non-destructive ultrasonic tests show that, for the compound casting plates poured at 1450 °C ± 10 °C, a permanent bond of the high-chromium steel with the grey cast iron was obtained within a contact surface area ranging from about 60% to 80%. As a result, these four compound castings were recognised as technologically useless. However, at a pouring temperature of 1550 °C ± 10 °C, a permanent bonding of both component materials was achieved, covering the entire contact surface. Therefore, the remaining tests presented herein refer only to these compound casting plates. The permanent bond within the compound casting plate confirms the cross-section macrostructure, whose example, cut from the centre of the bi-metal, is presented in Figure 6c.

In turn, Figure 7 presents a microstructure of the bond area within the compound casting plate. The structure of the bi-metal bond area is formed as a result of diffusion, first of all of C in the direction from the supporting part to the working part and cooling down from the high temperature to which the monolithic insert is heated up, and whose source is the liquid cast iron poured into the mould.

In general, in the bonding area of X46Cr13 steel with grey cast iron, two classical zones in terms of microstructure typical for bonded materials, i.e., zone 1(WL) corresponding to the microstructure of X46Cr13 and zone 5(SP) corresponding to the microstructure of grey cast iron, can be distinguished. The area of zone 1(WL) because of the distance from the cast iron part, is not carburised but only heated to a high temperature, ranging from about 1000 to about 1150 °C, as demonstrated in another author’s paper [19]. It should be noted that the mentioned temperature of the insert in zone 1(WL), i.e., exactly in the area from the bi-metal surface layer to the border with zone 2(WL`) (whose microstructure is formed as a result of both high temperature and carburisation), is within the range typical for the austenitising temperature used in the classic heat treatment processes of X46Cr13 alloy steel [25,26]. Therefore, for the X46Cr13 alloy steel, which shows the phenomenon of self-hardening, heating up to the phase stability temperature γ combined with cooling in the sand mould with an average rate of about 0.1 °C/s [19] resulted in obtaining a working part microstructure different from the initial state before the mould pouring, i.e., the initial microstructure composed of Cr(Fe)_23_C_6_ carbides into the ferritic matrix (Figure 8a), with a hardness of 200 μHV0.5, transforms to form a final microstructure composed of Cr(Fe)_23_C_6_ carbides in a martensitic-pearlite matrix (Figure 8b), with a hardness of 460 μHV0.5 (Figure 9). The phase composition of the presented microstructures of X46Cr13 steel in various states is typical, as described in detail in the papers [26,27,28]. In turn, zone 5(SP) has a microstructure typical for EN-GJL-HB255 grey cast iron, i.e., flake graphite in a pearlite matrix. 

Between zones 1(WL) and 5(SP), there are three transition zones. As a result of carburising and cooling from a high temperature, at a moderate speed, i.e., the phase transformations take place in the solid state, the narrow zone 2(WL`) is created. Since the cooling of the zone from a high temperature is carried out at a lower rate compared to the cooling rate for zone 1(WL), the dominating component of Cr(Fe) carbides is pearlite, which, in turn, results in reducing hardness to the level of 420 μHV0.5. 

Another zone 3(TZ) is formed from the liquid phase as a result of the carburisation of the border area of the steel insert by the cast iron, which lowers its liquidus and solidus temperature, and next by melting of this area as a result of the impact of the high temperature of the liquid cast iron. The chemical composition formed in this zone determines the crystallization of the microstructure with a final hardness of 540 μHV, composed of large amount of Cr(Fe) carbides, mainly M_7_C_3_ in a pearlite matrix (Figure 10, Figure 11 and Figure 12). 

Then the pearlite zone 4(SP`) appears, representing decarburised microstructure of the casting supporting part (Figure 13).

Figure 14 presents a view of three bi-metal plates with an average surface hardness of 450 HV5 ± 12 HV5 after use under the operating conditions of a coke quenching car. For those plates and three mottled cast iron control plates with an average surface hardness of 350 HV5 ± 14 HV5, operated in the same conditions, their weight and thickness were measured. It was found that the bi-metal plates showed both volumetric and mass loss at a level comparable to plates made in whole of mottled cast iron (Figure 15). 

More accurate observations showed that for compound castings, the highest-damage areas are located at the outer edges of the bi-metal working layer. For example, Figure 16 presents a view of a plate compound casting fragment after completing the operation cycle with visible, almost complete local loss of the X46Cr13 alloy steel working layer. The probable cause of the presented damage on the long edges of bi-metal plates is a local disconnection between the stainless steel and grey cast iron in a condition of multi-cycle thermal shock operation. This phenomenon generates an air gap between the working layer and the supporting part, increasing damage as a result of worsened heat conduction in the cyclically heated and cooled steel part and making it possible for water to ingress into the gap, increasing corrosion. However, it should be noted that the average mass loss is at the level of approx. 7%, while maintaining the bonding stability on almost the entire contact surface in this type of compound casting plates of alloy steel–grey cast iron determines its suitability for further use as coke quenching car lining. Further suitability of lining plates for use is an extremely important factor from the point of view of coke quenching car overhauls. Removing the need to dismantle individual plates, typically carried out by mechanical cutting of heavily corroded mounting screws in the overhead position, allows operators to reduce the costs and time of overhauls.

For mottled cast iron castings, the areas of increased damage occur locally, both in internal and external areas of the plate. In addition, despite the low average mass loss, not exceeding 6.5% on average, a crack was found in one of the plates under analysis (Figure 16). The cause of the presented damage by the thermal crack of a mottled cast iron plate is the non-uniform distribution of the carbon precipitations in the microstructure of the casting and the advantage in the amount of cementite (which increases the resistance on abrasive wear) to flake graphite (which increases heat conduction) in the pearlite matrix. The presence of a through-plate crack (Figure 17) makes it, unlike the remaining two, unsuitable for further use as a coke quenching car liner. It can be concluded that the through-plate crack of one of the three plates is a part of the trend that accompanies the use of this cast iron in coke quenching cars. According to our observations made during tests, between 30 and 40% of mottled cast iron plates are damaged by cracking before the entire car is withdrawn from service and sent for overhaul.

During the laboratory tests of coke quenching car lining plates, it was found that the coke production cycle resulted in slight carburising, sulphurising, and phosphating of the working layer of the bi-metal casting (Table 2). In addition, it was found that reducing the concentration of Cr by 0.2 wt.%, with the total concentration of this component exceeding 14 wt.%, can be considered negligible, e.g., from the point of bi-metal working layer corrosion resistance. Similarly, changes in the concentration of other components can be considered less significant.

However, based on the conducted metallographic tests (Figure 18), it was found that the 18,500 completed coke production cycles practically did not cause any significant changes in the individual transition zones of the compound casting. Only when compared to the initial state before starting the operation did the microstructure of the bi-metal working layer (zone 1(WL)) show a slightly increased amount of pearlite at the expense of martensite. As a result, the hardness of the working layer, compared to the initial state before starting the operation, is slightly reduced by 30 HV5, i.e., to the level of 420 HV5 ± 9 HV5. However, the scope of changes in the microstructure of the working layer and its hardness might be as well considered negligibly small, especially considering the quantity of heating and cooling cycles the test castings were subjected to. 

Summing up, the scope of the conducted tests covered the technology of creating compound castings composed of two parts, i.e., the working layer made of X46Cr13 high-chromium steel, and the supporting part made of EN-GJL-HB 255 grey cast iron, in a liquid–solid system by pre-installing a monolithic insert in the mould cavity.

The decisive impact of the C diffusion phenomenon, shown in the paper on forming the bond between the working layer and supporting layer materials, results obviously from the fact of easy transport of the component by an interstitial mechanism from one material to another, as opposed to the vacancy mechanism taking place, e.g., for Fe, Mn, Cr, Mo, or Ni. Omitting other components in the analysis, apart from C, present in the chemical composition of both the working layer and the supporting part is more justified, because despite the wide range of concentration variability, e.g., for chromium, the results of the conducted tests show no noticeable quantitative impact on the quality of bonding between both parts. 

Considering this, it was found that the lasting diffusion bonding between the component materials of compound castings, made in the arrangement alloy steel working layer–grey cast iron supporting part, is created primarily as a result of the transport of C towards the lower concentration, i.e., from the supporting part to the working layer. Since the material of the supporting part, made in the presented technology, is poured into a mould where the working part material is already placed in a form of a steel plate, the transport of carbon is associated, in the first stage, with diffusion in the liquid–solid system, and after the solidification process is completed, with the mechanism of diffusion in the solid state. Of course, the above-mentioned diffusion processes are heat-activated, and therefore the C transport is preceded by heat transport in the same direction, i.e., from the “hot” liquid alloy poured into the mould to the “cold” monolithic insert. Therefore, obtaining a permanent diffusion bonding over the entire contact surface between the working layer and a supporting part requires using a suitably high pouring temperature of cast iron.

The course of the presented processes allows us to conclude that creating a permanent diffusion bonding in the compound castings under analysis takes place in four successive stages:

**Stage I:** Wetting the monolithic insert with liquid cast iron poured into the mould. At this stage, it is necessary to meet the condition θ < 90° for the contact surface of the insert, where θ is the wetting angle formed by applying flux to the contact surfaces of the monolithic insert. 

**Stage II:** Heat-activated C diffusion in the direction from the liquid cast iron to the steel monolithic insert. At this stage, we apply a temperature and concentration condition for the liquid alloy–insert system:(1)$$\left\{\begin{array}{c} T_{\mathrm{SP}}>T_{\mathrm{WL}}\\ C_{\mathrm{SP}}>C_{\mathrm{WL}} \end{array}\right.$$

**Stage III:** Melting a thin layer (up to several dozen μm) on the surface of the monolithic insert remaining in direct contact with the liquid alloy as a result of carburising that reduces the temperature of liquidus and solidus. Hypothetical temperature and carbon concentration distribution in the bi-metal system under analysis at this stage is presented illustratively in Figure 19.

**Stage IV:** Solid state C diffusion. At this stage, the structure of the compound casting, formed mainly during stage III, is consolidated and meets the following concentration condition:(2)$$\left\{\begin{array}{c} C_{\mathrm{SP}}>C_{\mathrm{SP}`},C_{\mathrm{WL}`},C_{\mathrm{WL}}\\ C_{\mathrm{WL}}<C_{\mathrm{WL}`},C_{\mathrm{TZ}},C_{\mathrm{SP}`},C_{\mathrm{SP}}\\ C_{\mathrm{SP}}\approx C_{\mathrm{TZ}} \end{array}\right.$$

The presented phenomena of transporting mass and heat determine the formation of compound castings, where for the material arrangement grey cast iron–high-chromium steel, five basic zones can be distinguished. Successively, in the direction from the supporting part to the working layer, there are the following:A zone (SP) appropriate for grey cast iron, used for the supporting part containing flake graphite in a pearlite matrix;A pearlite zone (SP`) formed in the near-border area of the supporting part, created as a result of decarburisation associated with the transport of C towards a lower concentration area, i.e., to the working layer;A transition zone (TZ) composed of a large amount of Cr(Fe) carbides, mainly M_7_C_3_ in a pearlite matrix, formed from the liquid phase as a result of carburising reducing the melting point for the working layer border area and then melting it by the high temperature, and obtained as part of the heat transport from liquid alloy to the monolithic insert;A zone (WL`) formed in the working layer in the solid phase by carburising (insufficient to significantly lower the melting point) and cooling from the high temperature (insufficient to melt this area), determining the occurrence of microstructure with Cr(Fe) carbides in a pearlite-martensitic matrix with a dominant amount of pearlite;A zone (WL) with a microstructure formed as a result of self-hardening of X46Cr13 steel i.e., Cr(Fe) carbides, mainly M_23_C_6_ in a martensitic-pearlite matrix with a dominant amount of martensite.

In addition, analysing the results of tests conducted under the real coking plant operating conditions shows the suitability of compound casting plates in the arrangement X46Cr13 alloy steel–EN-GJL-HB 255 grey cast iron for use as coke quenching car liners. Such bi-metal plates are characterised by the stability of their microstructure and, consequently, their performance in cyclically variable heating and cooling procedures, necessary in the process of operating a coke quenching car. Considering this, thanks to their utility properties, such castings can successfully replace the plates used as liner plates for the coke quenching cars to date, made in whole of EN-GJN-HB 340 low-alloy mottled cast iron. It should be noted that the production cost, and consequently the purchase price, of bi-metal plate in the material arrangement under analysis is about 50% higher than for the full cast iron plate. The greatest impact on the cost of producing a bi-metal plate is the price of a 5 mm thick plate made of high-chromium steel and the cost of preparing a sand mould, considering the procedure for preparing a monolithic insert presented in the paper. However, this cost can be compensated by the higher service life of bi-metal plates compared to those made of cast iron, which results in no need to replace the damaged (cracked) plates. Thus, on one hand, savings are introduced due to the lack of the need to purchase a new plate to replace the damaged one, as already discussed, while on the other hand, it reduces the costs of coke quenching car overhauls as well.

## 4. Conclusions

Based on the analysis of test results, the following conclusions were elaborated:The phenomena that determine the formation of permanent bonding between the two connected alloys, being component materials of a compound casting, include heat and C transport in the direction from the “hot and high-carbon” material of the supporting part poured into the mould to the “cold and low-carbon” material of the working layer, placed as a monolithic insert inside the mould cavity. Therefore, the process of forming a permanent bonding between the working layer and supporting part in the presented technology of compound casting takes place successively as a result of wetting the monolithic insert contact surface with liquid alloy poured into the mould and then by thermally activated C diffusion in the liquid alloy–insert system.The most important aspect of forming a permanent diffusion bond between the two alloys in the presented compound cast iron technology was to create conditions for a transitional zone to emerge by carburising the material to reduce the temperature of the liquidus and solidus and then melting the thin layer of steel monolithic insert remaining in direct contact with the liquid cast iron poured into the mould.Using the compound casting in the arrangement X46Cr13 alloy steel–EN-GJL-HB 255 grey cast iron to replace the EN-GJN-HB 340 mottled cast iron currently used to cast lining plates for coke quenching cars allows us to extend the service life of the components, primarily due to their higher resistance to cracking.

## Figures and Tables

**Figure 1 materials-17-03539-f001:**
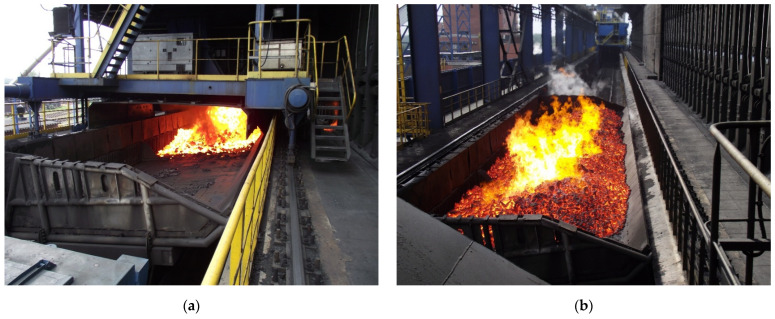
A view of the coke production cycle process: discharging heated coke pushed out from the bank of coke-ovens onto the surface of a quenching car (**a**), carrying the heated coke from the discharge site to the quenching tower (**b**), and the car in the quenching tower (**c**) with a subsequent wet coke quenching process (**d**).

**Figure 2 materials-17-03539-f002:**
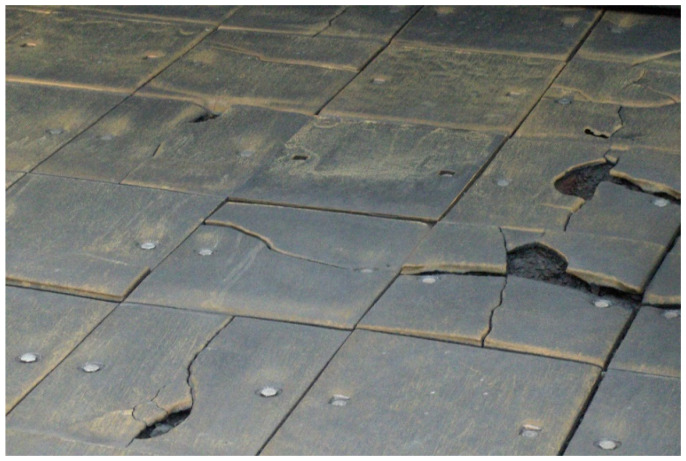
A view of damaged cast iron casting plates used as a lining for a coke quenching car, still operated in spite of the destruction.

**Figure 3 materials-17-03539-f003:**
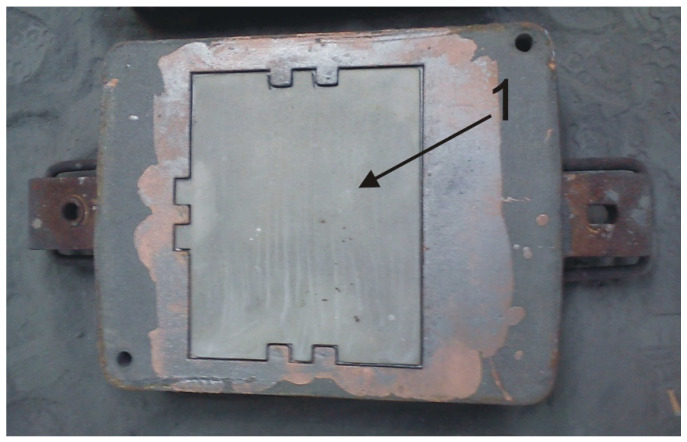
A view of the sand mould with a monolithic insert of X46Cr13 steel plate placed inside its cavity (1).

**Figure 4 materials-17-03539-f004:**
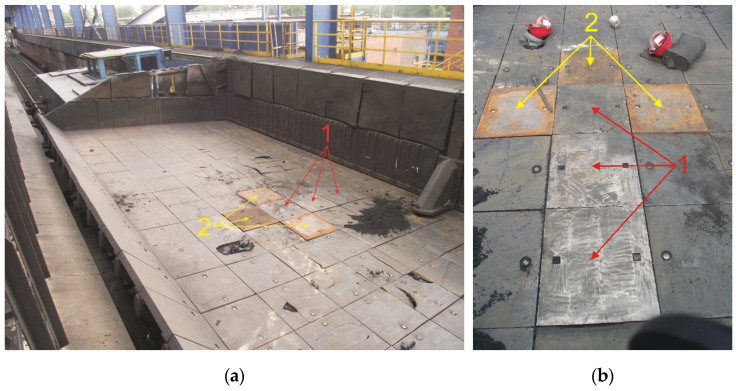
Installing 3 full-size compound castings of a X46Cr13 alloy steel working layer with a supporting part of EN-GJL-HB 255 (1) in a coke quenching car and 3 control EN-GJN-HB 340 mottled cast iron plates (2): (**a**) general view and (**b**) detailed view.

**Figure 5 materials-17-03539-f005:**
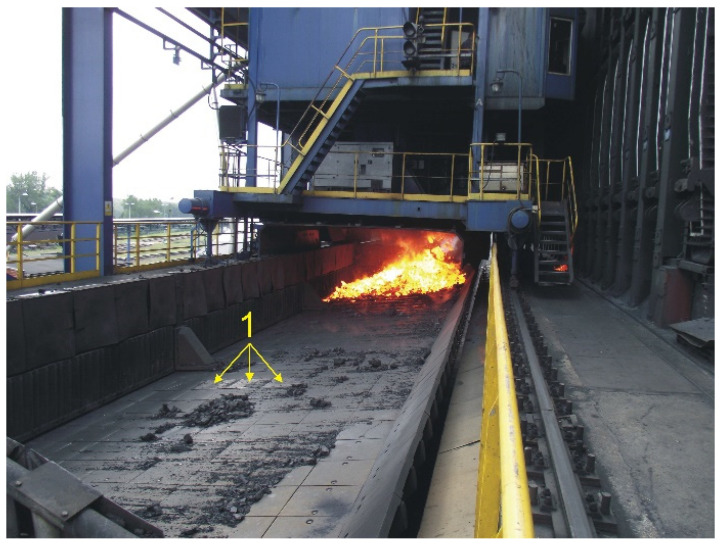
A view of plate compound castings of a X46Cr13 alloy steel working layer with a supporting part of EN-GJL-HB 255 grey cast iron (1) during the operation of a coke quenching car.

**Figure 6 materials-17-03539-f006:**
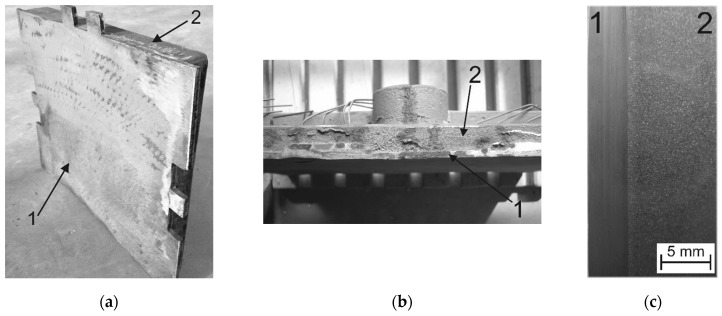
A view of a compound casting plate example of the system of X46Cr 13 (1) steel and EN-GJL-HB 255 (2) grey cast iron: (**a**) general view, (**b**) side view, (**c**) fragment of a macroscopic cross-section.

**Figure 7 materials-17-03539-f007:**
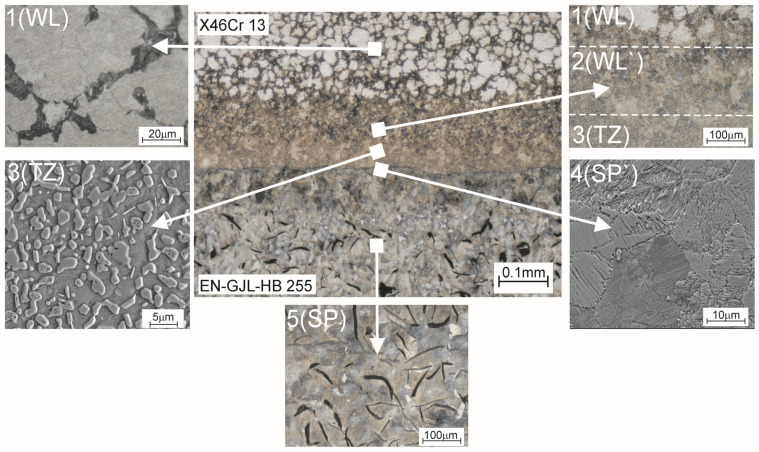
Microstructure of the bond area within the compound casting in the system: X46Cr13 steel working layer with the supporting part of EN-GJL-HB 255 grey cast iron.

**Figure 8 materials-17-03539-f008:**
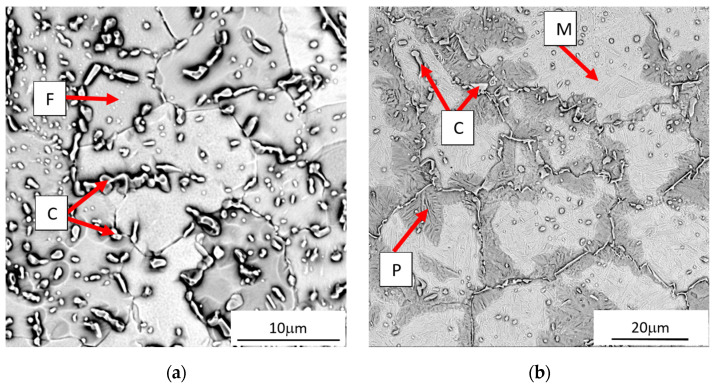
The microstructure of X46Cr13 steel (**a**) as-delivered monolithic insert (before the mould pouring) and (**b**) in the compound casting (after the pouring and cooling down of the mould); F—ferrite, C—Cr(Fe) carbides, P—pearlite, M—martensite.

**Figure 9 materials-17-03539-f009:**
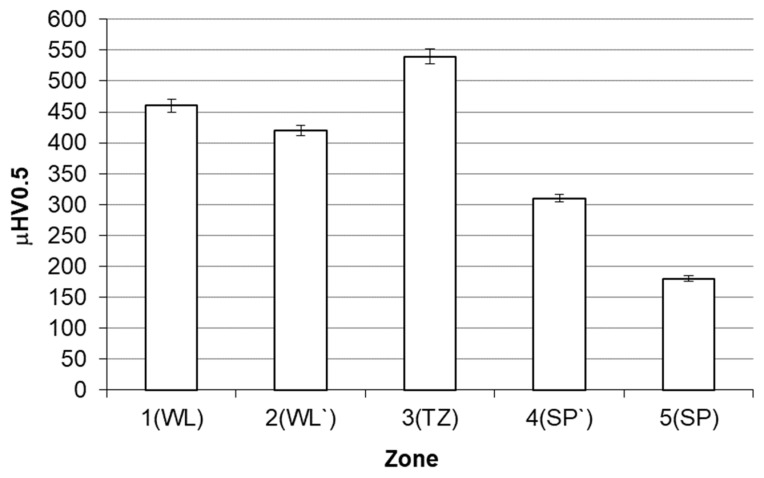
Distribution of hardness in individual zones of the compound casting: X46Cr13 steel–EN-GJL-HB 255 grey cast iron (zone marking as in Figure 7).

**Figure 10 materials-17-03539-f010:**
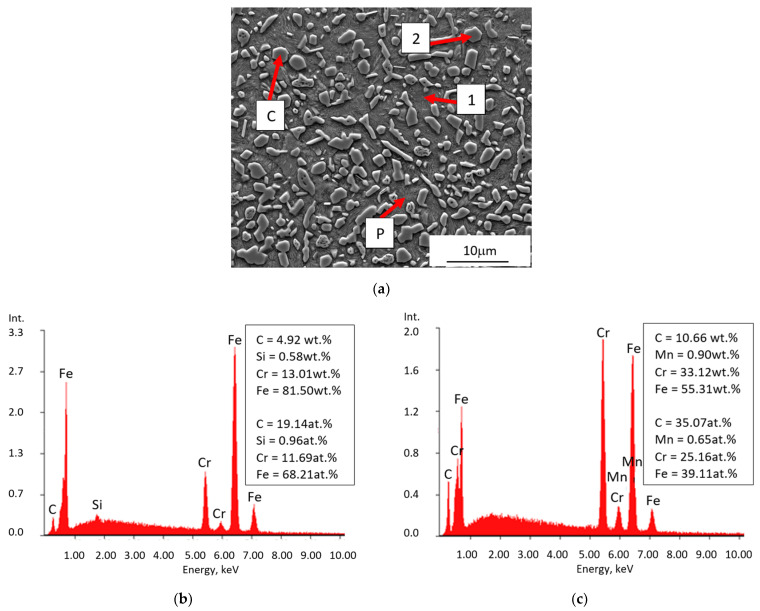
The microstructure of zone 3(TZ) in the compound casting of X46Cr13 steel–EN-GJL-HB 255 grey cast iron: (**a**) test area, (**b**) EDS results at point 1 from (**a**), (**c**) EDS results at point 2 from (**a**); P—pearlite, C—Cr(Fe) carbides.

**Figure 11 materials-17-03539-f011:**
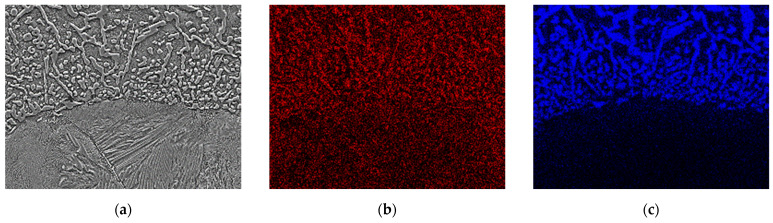
Surface distribution of elements on the border between zone 3(TZ) and 4(SP`) in the compound casting of X46Cr13 steel–EN-GJL-HB 255 grey cast iron: (**a**) test area, (**b**) C, (**c**) Cr, (**d**) Fe, (**e**) Si, and (**f**) Mn.

**Figure 12 materials-17-03539-f012:**
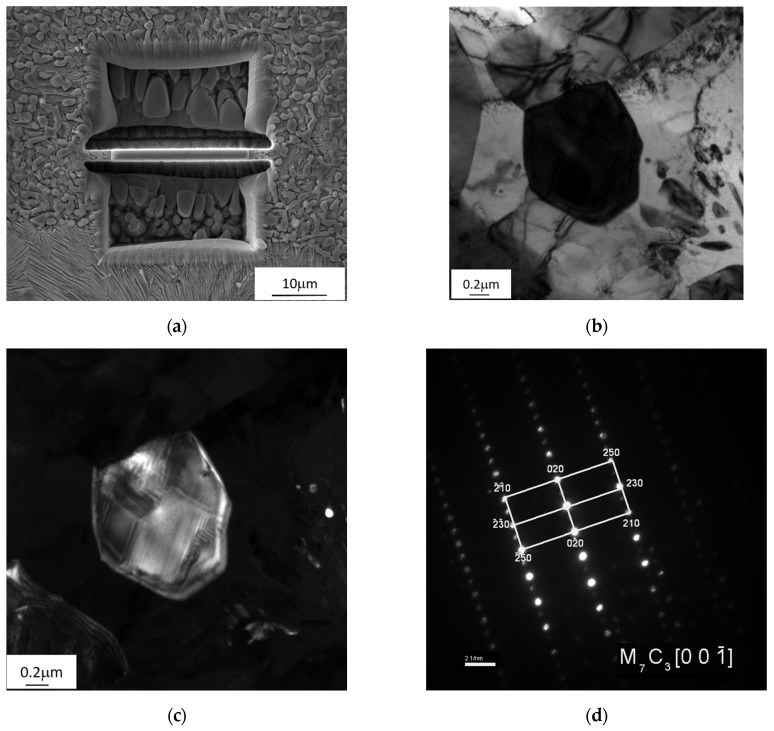
M_7_C_3_ carbide in pearlite in zone 3(TZ) in the compound casting of X46Cr13 steel–EN-GJL-HB 255 grey cast iron: (**a**) method of taking a lamella from zone 3(TZ) for TEM tests, (**b**) bright field, (**c**) dark field, and (**d**) selected area electron diffraction (SAD).

**Figure 13 materials-17-03539-f013:**
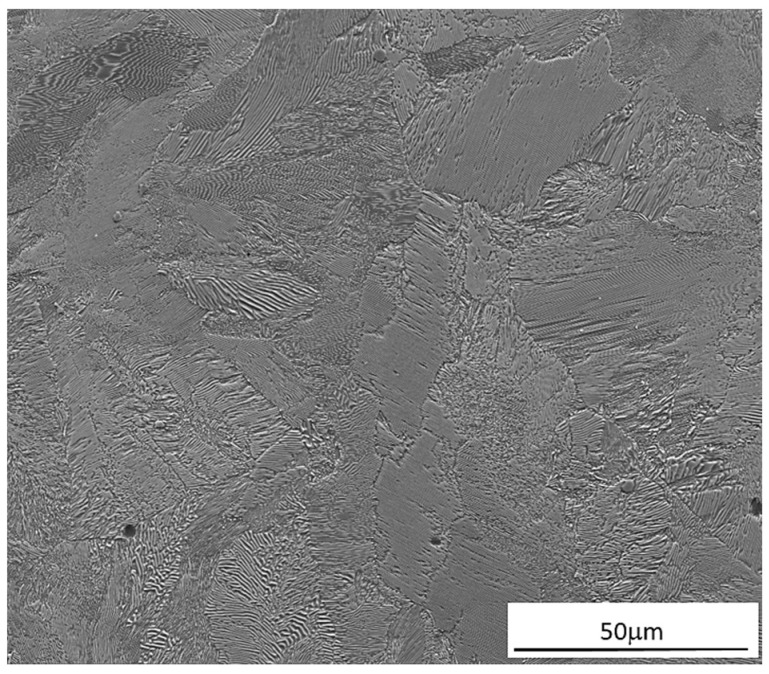
Pearlitic microstructure in zone 4(SP`) in the compound casting of X46Cr13 steel–EN-GJL-HB 255 grey cast iron.

**Figure 14 materials-17-03539-f014:**
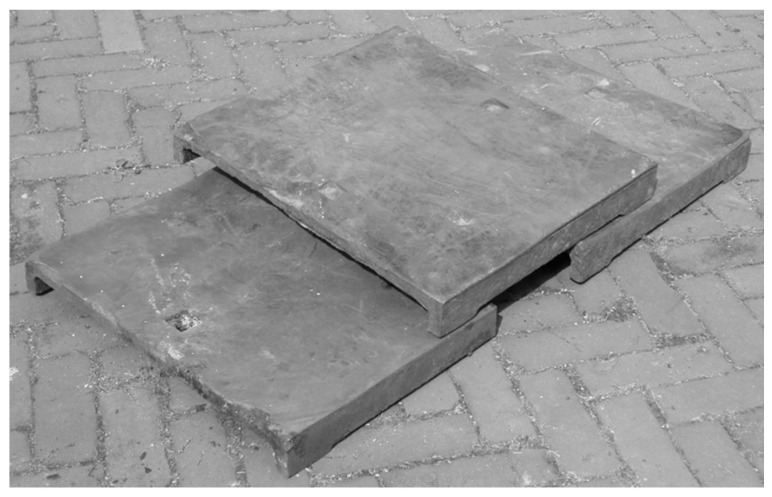
A view of three bi-metal plate castings designed for coke quenching car lining after completing 18,500 coke production cycles.

**Figure 15 materials-17-03539-f015:**
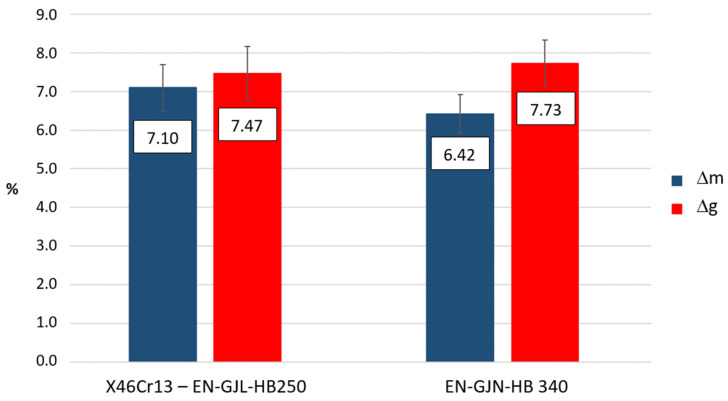
Average mass loss (Δm) and volumetric loss (Δg) of coke quenching car lining plates after completing 18,500 coke production cycles, carried out of compound castings of X46Cr13–EN-GJL-HB 255 and EN-GJN-HB 340 mottled cast iron.

**Figure 16 materials-17-03539-f016:**
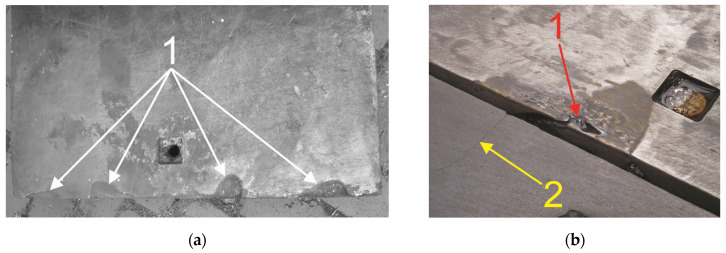
A view of the damage of a compound casting (1 in (**a**,**b**)) and a crack in an EN-GJN-HB 340 mottled cast iron casting (2 in (**b**)) lining plates after use under the operating conditions of a coke quenching car.

**Figure 17 materials-17-03539-f017:**
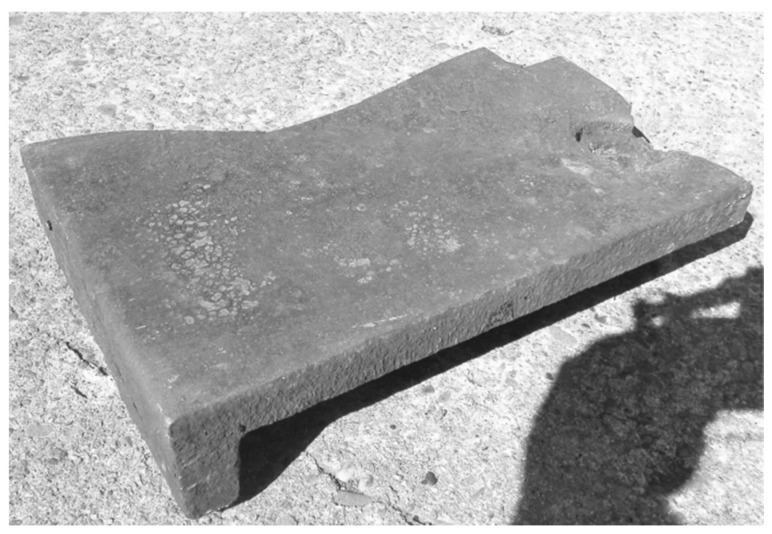
A view of a damaged test EN-GJN-HB 340 mottled cast iron test lining plate.

**Figure 18 materials-17-03539-f018:**
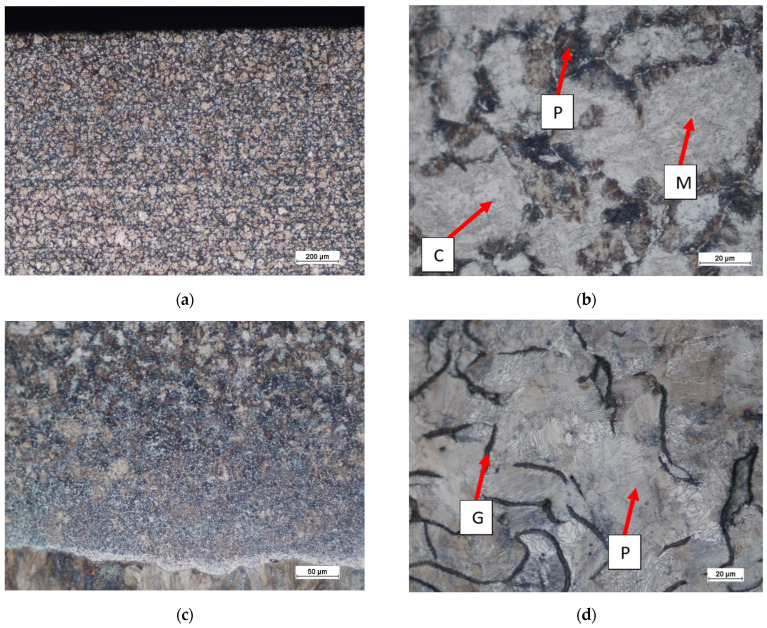
Microstructure of a lining plate compound casting of X46Cr 13–EN-GJL-HB 255 after use in a coke quenching car; (**a**,**b**) working layer (zone 1(WL); (**c**) area of bonding with diffusion transition zones 3(TS)—4(SP`); (**d**) supporting part (zone 5(SP); P—pearlite, M—martensite, C—Cr(Fe) carbides, G—flake graphite.

**Figure 19 materials-17-03539-f019:**
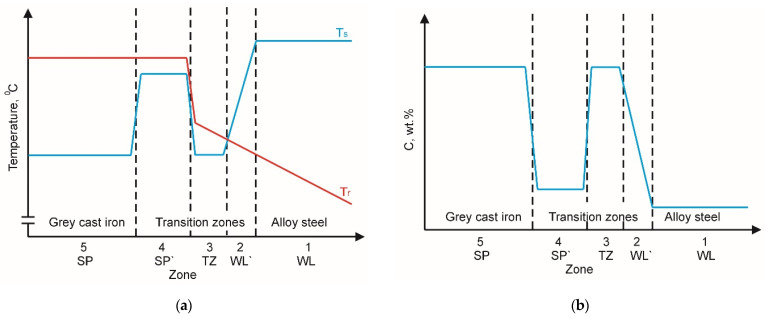
Hypothetical temperature (**a**) and C concentration (**b**) in individual zones of compound casting structure in the period corresponding to stage III: T_S_—solidus temperature T_r_—actual temperature of the bi-metal system under analysis.

**Table 1 materials-17-03539-t001:** Chemical composition of the tested alloys.

Elements Concentration, wt.%										
C	Mn	Si	Cr	Ni	Mo	Cu	Ti	V	S	P
X46Cr13										
0.440	0.600	0.340	14.500	0.650	0.090	0.001	0.001	0.001	0.001	0.020
EN-GJL-HB 255										
3.18	0.50	2.10	0.62	0.05	0.01	0.15	0.01	0.01	0.06	0.10
EN-GJN-HB 340										
3.07	0.79	1.54	1.60	0.06	0.01	0.04	0.01	0.01	0.02	0.04

**Table 2 materials-17-03539-t002:** Chemical composition of the X46Cr13 steel working layer in the plate compound casting after use in a coke quenching car.

Elements Concentration, wt.%										
C	Mn	Si	Cr	Ni	Mo	Cu	Ti	V	S	P
0.514	0.560	0.359	14.300	0.535	0.080	0.001	0.001	0.001	0.020	0.030
Difference from the state before operation										
+0.074	−0.040	+0.019	−0.200	−0.115	−0.010	-	-	-	+0.019	+0.010

## Data Availability

Data are contained within the article.

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
