# Peer review of "Compound Castings for the Coke Industry"

_materials, 2024, doi:10.3390/ma17143539_

Round 1

Reviewer 1 Report

Comments and Suggestions for Authors

1.       ntroduction: There are no references in the first two paragraphs.

2.       Figure 1 – 3 can be merged in one, so that the complete process is in one figure.

3.       Introduction: In several parts the existence of abrasive wear is mentioned. How does this wear occur, and do you only have abrasive wear or other wear mechanisms such as oxidative, adhesive also coexist?

4.       Introduction: I would recommend that you are also more specific on the corrosion phenomena that you refer too.

5.       Introduction: There is a lot of information given. Thus, I would suggest that in the end of the introduction you added a concluding paragraph clearly stating the research aim, novelty and main outcome of this work.

6.       Materials and methods: The reason for selecting two casting temperatures should be mentioned.

7.       Materials and methods: I would recommend that you mentioned the location in te ingot from which the samples were taken, as microstructure varies in castings depending on solidification area.

8.       Materials and methods: How do you know that the weight loss is only due to wear? Also when referring to wear measurements several factors should be described e.g. tribo-system, condition, motion, cycles etc.

9.       Materials and methods: Number of repeats for hardness measurements should be mentioned.

10.   Results: You mention that C diffusion took place from supporting part to working part. Is this comment based on microstructural changes or did you try to quantify carbon content?  

11.   Results: How did you control and measure the solidification rate?

12.   Results: Range of experimental values should be included.

13.   Error bars should be added in Figure 9.

14.   Results: Suitability tests for industrial applications should be move to materials and methods.

15.   Results: I would remove the term wear, because it purely refers to tribological phenomena. In this case material loss can be due to a combination of phenomena e.g. thermal shocks, mechanical loads, corrosion, delamination etc. Also, the images showing failure are too macroscopic to identify and conclude on a mechanism.

16.   The discussion is rather limited and is more like a brief summary of the results resented previously. I would suggest that you either merge it to the results (Results and discussion) or expanding it by strengthening the theoretical background and justifying results and mechanisms.

Author Response

Dear Sir,

thank you for Your review of our manuscript materials-3084025 entitled „ Compound Castings for the Coke Industry”. All Your comments are addressed in the attached PDF file as suitable responses for Reviewer #1. The necessary modifications made in the manuscript are marked in green text.

Best Regards

Author of paper

Reviewer 2 Report

Comments and Suggestions for Authors

In my opinion authors should modify/clarify the following issues:

#1) Page 1. Line 37, 43. Please add references in the beginning of the introduccion section. The two first paragrahs have no references.

#2) Page 2. Lines 49-51 Please do not use lumping references. Please cite each one individually describing the main achivements of such research. Please do the same hereafter in the manuscript. For instance in line 54.

#3) Page  2. Line 68. Please delete “and”, please use “[9,10]” instead.

#4) In my opinion the introduction section is too long and it should be reduced including a brief description of the aims of the study based on the research gap. In addition the description of the method (lines 72-106), in my opinion should be included in section 2. Materials and methods.

#4) Page 4. Line 107. Section 2 is presentes as a list of bullet points. In my opinion this is not right and I suggest authors to rewrite this section in a more conventional way. In addition, in my opinion methods are not properly described and authors should include more information. For instance, page 5. Line  “checking the chemical composition”, how this was done?  Please include more details. Line 143, show the test lay out. Do the same for the other points.

#5) Page 8. Lines 233-234. The Figures 10 -12 and 13 are not properly discussed in the paper. Please include a description and analysis of the results shown in such figures.  The size of the labels of x-axis and y-axis of Fig. 10 are too small.

#6) page 12. Line 282. Please be more precise, please explain how the avergae mass loss is estimated as 7%. Do the same in line 289.

#7) Page 15. Line 386. The Figure 20 caption is wrong it should be Figure 21. In addition, in my opinion the graphs in Fig. 21 are not properly discussed. Please correct this.

#8) Reference section. Please merge reference #21 and #22. In addition please include more recent papers since only 4 out of 22 are published in the last 5 years.

Comments on the Quality of English Language

Minor editing of English language required

Author Response

Dear Sir,

thank you for Your review of our manuscript materials-3084025 entitled „ Compound Castings for the Coke Industry”. All Your comments are addressed in the attached PDF file as suitable responses for Reviewer #2. The necessary modifications made in the manuscript are marked in blue text.

Best Regards

Author of paper

Reviewer 3 Report

Comments and Suggestions for Authors

Author presents results on new technology for compound casting for the coke quenching car. It is stated that the plates, produced by this new technology, were mainly damaged at the outer edge of the bimetal working layer (figure 18). A few explanations are needed here. Were the new plates surrounded by the classically produced plates or the same new ones? That is, which edge of the new plates was damaged? To be precise, did the author consider influence of the neighboring plates on the edge damages?

The same additional explanation is needed for the plate that was fully cracked.

Does the author plan to test the quenching car with all the newly produced plates? Then, the “influence” of the neighboring plates would be the same at all the edges.

The last point in discussion (lines 402 to 409) on lower costs for producing the new plates is not supported by explicit numbers. It is stated that the manufacturing costs of the new plates are 50 % higher than the costs of full cast iron plates production. How long should the service life of the new plates be extended to cover for the increased costs of their manufacturing?

Another point is that there is no estimate how long can the new plates, when “sightly” damaged still be operational, i.e. what is the risk for operational safety of the quenching car?

What concerns the style of presentation, there are only a few minor points. Do not use the phrases that assume the article “doing” something, like in the first sentence of the Abstract (“The paper presents…”). Please, write this in a neutral form, article cannot “do” anything, author does.

The phrase “picture taken by the author” that appears in captions of a few figures is not necessary. It is understood that the author has produced all the results, including figures, diagrams, calculations, etc. i.e., if another reference [#] was not given.

The scanned pages of the manuscript, with marked errors and suggested corrections, are eclosed.

Comments on the Quality of English Language

Author Response

Dear Sir,

thank you for Your review of our manuscript materials-3084025 entitled „ Compound Castings for the Coke Industry”. All Your comments are addressed in the attached PDF file as suitable responses for Reviewer #3. The necessary modifications made in the manuscript are marked in yellow text.

Best Regards

Author of paper

Round 2

Reviewer 1 Report

Comments and Suggestions for Authors

Dear author,

After reading the updated version of the manuscript and your point-by-point reply to my comments, I now believe that this work is appropriate for publication.

Comments on the Quality of English Language

Minor editing of English language required